# The Effects of Exercise Habit on Albuminuria and Metabolic Indices in Patients with Type 2 Diabetes Mellitus: A Cross-Sectional Study

**DOI:** 10.3390/medicina58050577

**Published:** 2022-04-23

**Authors:** Hsin-Yi Kuo, Ya-Hui Huang, Su-Wen Wu, Feng-Hsun Chang, Yi-Wei Tsuei, Hsin-Chiung Fan, Wen-Fang Chiang, Po-Jen Hsiao

**Affiliations:** 1Department of Nursing, Taoyuan Armed Forces General Hospital Hsinchu Branch, Hsinchu 300, Taiwan; singeiyo@yahoo.com.tw (H.-Y.K.); y11260@yahoo.com.tw (Y.-H.H.); vivian63326@yahoo.com.tw (S.-W.W.); vn512560@msn.com (F.-H.C.); 2Department of Emergency Medicine, Taoyuan Armed Forces General Hospital, Taoyuan 325, Taiwan; er03@aftygh.gov.tw; 3Department of Nursing, Taoyuan Armed Forces General Hospital, Taoyuan 325, Taiwan; fan2667@yahoo.com.tw; 4Division of Nephrology, Department of Internal Medicine, Tri-Service General Hospital, National Defense Medical Center, Taipei 114, Taiwan; wfc96076@yahoo.com.tw; 5Division of Nephrology, Department of Internal Medicine, Taoyuan Armed Forces General Hospital, Taoyuan 325, Taiwan; 6Department of Life Sciences, National Central University, Taoyuan 320, Taiwan; 7Division of Nephrology, Department of Medicine, Fu-Jen Catholic University Hospital, School of Medicine, Fu-Jen Catholic University, New Taipei City 242, Taiwan

**Keywords:** diabetes mellitus, exercise habit, albuminuria, metabolic indices, blood glucose, plasma lipid

## Abstract

*Background and Objectives:* Diabetes mellitus (DM) can cause macrovascular and microvascular complications, potentially resulting in further life-threatening complications. In general, the global prevalence of type 2 DM is increasing. To date, the care of DM comprises three aspects: diet, medication and exercise; among them, exercise is the most economical. Albuminuria is associated with renal injury and the progress of chronic kidney disease (CKD). The effects of habitual exercise in patients with new onset of diabetic kidney disease (DKD) have not been generally recognized. Our aim was to conduct an observational study regarding the effects of regular exercise on proteinuria and associated metabolic indices in patients with newly diagnosed type 2 DM. To investigate the effects of an exercise habit on albuminuria and the metabolic indices including renal function, blood glucose, and plasma lipids among patients with newly diagnosed type 2 DM. *Materials and Methods:* A cross-sectional study was conducted on newly diagnosed DM patients in two teaching hospitals in Taiwan from 1 June to 31 December 2020. The DM patients participated in the Diabetes Shared Care Network. According to the DM care mode, the patients’ blood biochemical results were analysed. Based on exercise duration, the patients were divided into two groups, i.e., the exercise group (≥150 min per week) and the non-exercise group (<150 min per week). Clinical demographic features and laboratory examination including blood and urine biochemistries were determined. *Results:* A total of 229 patients including 99 males (43.2%) and 130 females (56.8%) participated in the study. The proportion of DM patients with normoalbuminuria was higher (*p* < 0.05) in the exercise group (69.8%) than in the non-exercise group (53.7%), and the proportion of DM patients with micro or macroalbuminuria was lower in the exercise group (30.2%) than in the non-exercise group (46.3%). Levels of glycated hemoglobin (HbA1c), fasting plasma glucose (FPG), triglycerides (TG) and high-density lipoprotein (HDL) were significantly different in both groups. Compared with the non-exercise group, lower HbA1c (6.89 ± 0.69 vs. 7.16 ± 1.05%) (*p* < 0.05), lower FPG (121.9 ± 25.7 vs. 140.5 ± 42.4 mg/dL) (*p* < 0.05), lower TG (115.6 ± 53.6 vs. 150.2 ± 15.4 mg/dL) (*p* < 0.05), and higher HDL (50.3 ± 11.4 vs. 44.1 ± 9.26 mg/dL) (*p* < 0.05) levels were noted in the exercise group. *Conclusions:* Regular exercise remains imperative and may bear an impact on albuminuria, blood glucose, and plasma lipids among type 2 DM patients. Therefore, medical staff and healthcare providers should encourage patients to maintain an exercise duration ≥150 min per week for preventing and controlling DM progression.

## 1. Introduction

Diabetes mellitus (DM) represents one of the fastest increasing chronic diseases globally. The number of DM patients has quadrupled in the past three decades, and DM is the ninth major cause of death globally. DM can result in high complication and mortality rates, resulting in a heavy burden of medical expenses [1,2,3]. To date, about 1 in 11 adults worldwide have DM, 90% of whom have type 2 DM. Asia is a major area of the rapidly emerging type 2 DM global epidemic [2,3,4,5]. Even genetic predisposition could partly contribute toward individual susceptibility to type 2 DM. The unhealthy diet and sedentary lifestyle may play an important role in the current global epidemic [6,7,8]. DM is a metabolic disease and associated with chronic inflammation in a complex immunological process. Insulin resistance (IR) can also cause a series of immune responses that aggravate the inflammatory state, further resulting in hyperglycemia [9]. Diabetic kidney disease (DKD) is one of the most common complications of DM. The pathological mechanism behind DM is insufficient insulin secretion (damage to the number and/or function of β cells) and IR in liver, muscle and fat cells, which can increase blood glucose levels and affect lipid metabolism, resulting in increases in cholesterol and triglycerides (TG) [10,11], which further leads to arteriosclerosis and cardiovascular disease [12]. Increases in blood glucose can also cause thickening or nodular sclerosis of the glomerular basement membrane (GBM), resulting in an increase in creatinine (Cr) and the production of proteinuria, which affects renal function [13]. At present, the three main measures for the treatment and care of DM are medication, diet and exercise; of these, exercise is the most economical. Regular exercise can reduce IR, increase insulin sensitivity and improve the efficiency of glucose entry into cells during exercise to reduce the concentration of glycated hemoglobin (HbA1c) and improve blood glucose levels [14]. In addition, exercise can improve plasma lipid levels, especially by reducing TG and increasing high-density lipoprotein (HDL) [15]. The studies of Stensvold et al. [16] and Larose et al. [17] also found that after interventions of aerobic, resistance and combined aerobic-resistance exercise, HbA1c, TG and HDL were significantly improved in the intervention group compared with the control group. Exercise could make an improvement in physical functioning, prevention of cardiovascular complications, delay the progression of renal dysfunction and the occurrence of proteinuria [18,19].

Overall, regular exercise was planned and repeated to improve or maintain physical health parameters in study participants. Exercising habitually and frequently helps in the prevention of serious illness such as cardiovascular disease, heart disease, metabolic disease, type 2 DM, and obesity. Many cases of type 2 DM are preventable with lifestyle modifications, such as maintaining a healthy body weight, adhering to a healthy diet, maintaining physical activity, and abstaining from smoking and alcohol consumption. The aim of our present work is to explore the association between habitual exercise and albuminuria and metabolic indices in patients with newly diagnosed type 2 DM.

## 2. Materials and Methods

### 2.1. Study Subjects and Locations

A cross-sectional study was conducted by the sampling of newly diagnosed type 2 DM patients from two teaching hospitals (Taoyuan Armed Forces General Hospital and Hsin Chu Armed Hospital) in Taiwan from 1 June to 31 December 2020. All patients were participants in the Diabetes Shared Care Network in Taiwan. Demographic features, biochemical data including available blood and spot urine sample results were investigated. Nephrologists, dietitians, and nurses were involved in this study. Only newly diagnosed type 2 DM patients in the educational program for DM and CKD with a fixed diet regimen of at least one month were included. Patients with confirmed diagnosis or clinical history of periodic paralysis, thyroid or adrenal disorders, inherited kidney disease, and renal tubular acidosis were excluded from the study. These comorbidity factors and associated diseases may have potential effects on proteinuria or albuminuria. To avoid other potential confounders and to focus on the target population of stable newly diagnosed type 2 DM patients, we additionally excluded patients with a history of hypertension, acute kidney injury, massive hematuria, renal transplant, dialysis treatment, bladder irrigation, prior creation of a neobladder, pregnancy, obstructive uropathy, and patients under 18 years of age. To avoid any confounding effects of medication and to solely investigate the original effects from the exercise habit, patients who had used any drugs including traditional Chinese medicine were also excluded in this study. For optimal health, it is recommended that adults perform at least 150 min per week of moderate-intensity exercise. In this study, our nurses asked all the participating patients to complete a self-report of exercise habits according to the US Physical Activity Guidelines. Based on the exercise habits, the patients were divided into two groups, i.e., the exercise group (≥150 min per week for more than 6 months) and the non-exercise group (<150 min per week for more than 6 months) [6,10,14]. The detailed report of exercise type revealed jogging as the most common among all subjects.

### 2.2. Research Tools

The analysis of medical records was conducted on a database of patients who underwent follow-up. The analysis examined the following factors: (1) Physiological data: sex, age, blood pressure (BP) and renal function stage for diabetic nephropathy (DN). (2) Test items: (1) DM control indicators: HbA1c and fasting plasma glucose (FPG); (2) plasma lipids: total cholesterol, TG, HDL and low-density lipoprotein (LDL); and (3) renal function: Cr and eGFR. In addition, urine albumin levels were determined based on regular urine collection. Urinary albumin excretion < 20 mg/L defines normoalbuminuria, 20–200 mg/L indicates microalbuminuria and >200 mg/L indicates macroalbuminuria. The urine protein-creatinine ratio (UPCR) and urine albumin-to-creatinine ratio (UACR) were then calculated. (3) To provide patients with self-management guidance and record patients’ daily physical activities. Only newly diagnosed stable DM patients who were not on medication with a fixed diet regimen in the educational program were included. To avoid other potential confounders and to focus on the target population, we also excluded patients with acute kidney injury, massive hematuria, renal transplant, pregnancy, obstructive uropathy, and under the age of 18.

### 2.3. Sample Size and Statistical Methods

Patients satisfying the inclusion criteria participated in the study, and the basic characteristics of demographic variables were analysed according to category. We designed a sample size of at least 100 for each group, calculated to provide 85.4%, 80% and 99.9%, 99.5% and 97.1% of power (alpha = 0.05, two-tail) on age, HbA1c and FPG, triglyceride, and HDL to detect statistically significant differences between the two groups [20]. Using the Kolmogorov–Smirnov test, no significant *p* values (0.200 and 0.052) were noted in both exercise and non-exercise groups, respectively, which confirmed the normality of distribution of HbA1c.

### 2.4. Data Analysis

After completion of data collection, SPSS for Windows 20.0 software was used for the statistical analysis. Descriptive statistics were used to describe the sex, age, renal function stage for DN and exercise distribution of the patients. HbA1c, FPG, total cholesterol, TG, HDL, LDL, Cr, eGFR, UPCR, UACR, urine albumin, BP and other indicators are expressed as mean ± standard deviation (SD), and independent t-test was used to determine the differences between the exercise and non-exercise groups.

## 3. Results

A total of 240 patients were initially included in the study. To avoid potential confounding effects of drugs, we subsequently excluded four and seven patients due to the use of anti-hypertensive drugs and oral hypoglycemic agents from the study. A total of 229 patients completed the study, including 99 males (43.2%) and 130 females (56.8%) with an average age of 63.41 ± 11.81 years. A total of 106 patients comprised the exercise group (46.3%), and 123 patients comprised the non-exercise group (53.7%) (Table 1). 

The levels of HbA1c, FPG, TG and HDL were all significantly different in both groups. Compared with the patients in the non-exercise group, the patients in the exercise group had lower HbA1c (6.89 ± 0.69 vs. 7.16 ± 1.05%) (*p* < 0.05), lower FPG (121.9 ± 25.7 vs. 140.5 ± 42.4 mg/dL) (*p* < 0.05), lower TG (115.6 ± 53.6 vs. 150.2 ± 15.4 mg/dL) (*p* < 0.05), and higher HDL (50.3 ± 11.4 vs. 44.1 ± 9.26 mg/dL) (*p* < 0.05) levels (Table 1).

This study found that patients with normoalbuminuria predominated in both groups, but a higher percentage of patients with normoalbuminuria in the exercise group than in the non-exercise group (69.8% vs. 53.7%) was noted. Lower percentages of patients with microalbuminuria and macroalbuminuria were included in the exercise group than in the non-exercise group (21.7% vs. 30.9% and 8.5% vs. 15.4%) (Table 2). Compared with those in the non-exercise group, lower percentages of patients with micro- or macroalbuminuria were observed in the exercise group (Figure 1). 

In terms of the severity of albuminuria, the analysis of biochemical results showed that albuminuria was significantly correlated with HbA1c, FPG, TG and HDL (Table 3). After multiple logistic regression analysis using exercise as a dependent variable, the analysis results also demonstrated a trend where albuminuria was strongly correlated with HbA1c, TG and HDL (Table 4). Statistical analysis was conducted using the general linear model (GLM) with regression coefficients, with the results shown in the Appendix A.

## 4. Discussion

Our study collected data from 229 newly diagnosed type 2 DM patients in Taiwan and found that HbA1c, TG and HDL levels were significantly different in the exercise group compared with the non-exercise group. Therefore, an exercise duration ≥ 150 min per week is speculated to delay the occurrence of albuminuria and reduce the risk of DN in DM patients.

Regular exercise is an important aspect of a healthy lifestyle as it can reduce cardiovascular disease, DM, and malignancy and is associated with decreasing hypertension, overweight, and obesity. In previous studies, Stensvold et al. [16] and Larose et al. [17] reported that the interventions of aerobic, resistance and combined aerobic-resistance exercise may significantly improve levels of HbA1c, TG and HDL in patients with metabolic syndrome. Research by Lin et al. [21] indicated that exercise could reduce total cholesterol and LDL, while in this study, total cholesterol and LDL were not significantly improved. According to the literature, DM patients should consider the type, intensity, frequency, and duration of exercise. Types of exercise include the following: (1) aerobic exercise, (2) muscular endurance exercise, (3) stretching exercises and (4) balance exercises. The exercise intensity should be low or moderate. Both the ratings of perceived exertion (RPE) and the targeted heart rate (THR) = maximum heart rate (220-age) × 50–70% recommended by the *2018 American Diabetes Association (ADA): Standards of Medical Care* can be used as references, and the most suitable exercise intensity for DM patients is that at which the patient can talk freely but cannot sing freely (i.e., an intensity that causes them to breathe heavily) during exercise. The recommended exercise frequency is 150 min per week. Exercise should take place within 1–2 h after a meal and can be performed multiple times per day, and each exercise session should be longer than 20 min [22,23,24].

Although the exercise duration was more than 150 min per week, whether the exercise intensity met the RPE recommendations of the 2022 ADA Standards of Medical Care was not evaluated in our study [24]. In addition, muscular endurance exercises with dumbbells, extendible ropes and resistance bands, stretching exercises and balance exercises, such as walking in a straight line, can be implemented to reduce joint stiffness and the risk of falls. Patients with CKD could be advised to increase their physical activity when appropriate [18]. Previous studies recommended that exercise can improve BP [25,26], but our present study showed no significant difference in BP between the two groups. In addition to exercise, diet and medication also affect BP. The BP of the two groups was controlled at 130 ± 1.3/80 ± 1.1 mmHg, which is close to the ideal BP range.

Renal function indicators, such as Cr, the eGFR, UPCR and UACR, also showed no significant differences between the two groups. According to the literature, the following factors affect renal function: (A) Age: the eGFR declines with age; (B) BP: if BP is not controlled to below 130/80 mmHg, it can cause glomerulosclerosis and affect renal function; (C) Blood glucose: the advanced glycation end-products (AGEs) and cytokines produced by hyperglycemia (HbA1c above 7%) can cause mesangial expansion and reduce the eGFR; (D) Plasma lipids, especially LDL: if LDL is not controlled to below 100 mg/dL, renal atherosclerosis, which affects renal function, can easily occur [13,27]. 

Among the indicators of albuminuria, microalbuminuria is a diagnostic indicator and an early clinical manifestation of DN. DN is the most common chronic complication of DM and one of the major causes of end-stage renal failure. Therefore, prevention of the progression of DM to DN is an important clinical task. Urinary albumin excretion, i.e., albuminuria, is a commonly used diagnostic criterion for DN. Albuminuria is divided into normoalbuminuria (urine albumin < 20 mg/L), microalbuminuria (urine albumin 20–200 mg/L), and macroalbuminuria (urine albumin > 200 mg/L) [3,4,5]. Proteinuria or albuminuria is also an imperative marker of renal function estimation, which can help in the early detection of kidney disease progression [28]. As well as being a major indication of kidney disease, albuminuria constitutes a marker of cardiovascular disease and kidney disease progression in addition to a predictor of mortality. A reduction in the degree of albuminuria has been demonstrated to improve both cardiovascular and renal outcomes [29,30]. In our study, the patients were stratified by exercise duration.

Exercise training has been recommended for patients with CKD by the Kidney Disease Improving Global Outcomes [31]. A substantial number of meta-analyses have confirmed the positive impacts of regular exercise programs for CKD patients on physical performance, cardiopulmonary function, blood lipids, and quality of life [32,33,34]. Previous review suggested that high levels of physical activity appeared to be closely related to low proteinuria [35], and an observational study on non-diabetic women revealed similar results [36]. Afshinnia et al. [37] reported that exercise training can reduce proteinuria in obese individuals, though the long-term effects have not been evaluated with high-quality experimental studies. However, the sedentary time of CKD patients, especially those with severe renal function impairment, is still significantly higher than that of individuals without CKD. Glavinovic et al. [38] reported that the sedentary time of CKD was 10-times higher than that of individuals without CKD. Based on the results of our study, in clinical practice, health education provided by medical staff to DM patients should include the following: (1) The exercise duration should be ≥150 min; (2) A self-assessment of exercise intensity can be considered as bearing the ability to talk freely but not to sing freely (i.e., breathing heavily) during exercise. With exercise that meets these indicators, blood glucose, TG and LDL can be reduced, HDL can be increased, and renal function can be maintained.

These results of our study can serve as a basis for future research on the exercise habits of type 2 DM patients and as a reference for medical staff to educate patients in clinical practice to promote patients’ health and improve their quality of life. However, our study has several limitations: the study results may have been influenced by several physiologic factors including age, gender, nutrition, exercise type (walking, hiking, swimming, etc.), lifestyle, diet, and various genetic conditions. In addition, we did not collect the 24-h urine samples to assess urinary albumin excretion in this study. In addition to the cross-sectional design, the need for further experimental study regarding the proposed evaluation, the lack of large sample size calculation and restrictions related to the statistical analysis represent the limitations in this study.

## 5. Conclusions

This study investigated the effects of exercise duration on the albuminuria and blood biochemistry results among newly diagnosed type 2 DM patients. The results showed that DM patients with exercise duration ≥150 min per week may have lower albuminuria, FPG, HbA1c, TG levels and higher HDL levels compared with the patients with exercise duration <150 min per week. Regular exercise remains vital for controlling DM. In clinical practice, healthcare professionals and physicians would encourage patients to maintain a habit of exercising ≥150 min per week during the coronavirus disease 2019 (COVID-19) pandemic.

## Figures and Tables

**Figure 1 medicina-58-00577-f001:**
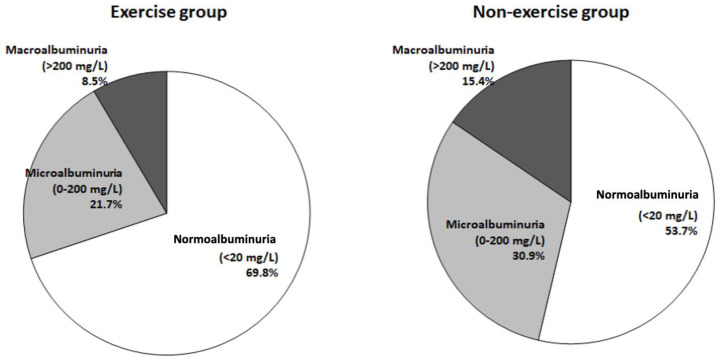
Percentages and levels of albuminuria between the exercise and non-exercise groups.

**Table 1 medicina-58-00577-t001:** Demographic and biochemical variables in both groups.

Variables	Exercise Group (*n* = 106)	Non-Exercise Group (*n* = 123)	*p*
Gender			0.931
Female	61 (57.5)	69 (56.1)	
Male	45 (42.5)	54 (43.9)	
Age (years)	66.2 ± 10.1	61.9 ± 12.8	<0.05 **
HbA1c (%)	6.89± 0.69	7.16 ± 1.05	<0.05 **
FPG (mg/dL)	121.9 ± 25.7	140.5 ± 42.4	<0.05 **
Cholesterol (mg/dL)	158.3 ± 27.2	158.5 ± 33.5	0.967
TG (mg/dL)	115.6 ± 53.6	150.2 ± 15.4	<0.05 **
Creatinine (mg/dL)	0.81 ± 0.19	0.79 ± 0.19	0.628
Uric acid (mg/dL)	5.59 ± 1.35	5.85 ± 1.75	0.205
HDL (mg/dL)	50.3 ± 11.4	44.1 ± 9.26	<0.05 **
LDL (mg/dL)	80.4 ± 21.6	84.1 ± 25.3	0.313
UPCR (mg/gm)	199.7 ± 36.4	244.6 ± 34.2	0.371
UACR (mg/gm)	7.66 ± 2.54	11.4 ± 2.41	0.285
SBP (mmHg)	130.9 ± 1.36	130.8 ± 1.31	0.955
DBP (mmHg)	76.8 ± 1.08	78.5 ± 1.18	0.287
Albuminuria (mg/L)	63.6 ± 196.0	112.0 ± 291.3	0.139

Values are expressed as numbers (percentages) or the means ± standard deviations. ** Difference between the exercise and non-exercise groups (*p* < 0.05). Abbreviations: eGFR: estimated glomerular filtration rate; FPG: fasting plasma glucose; HDL: high-density lipoprotein; LDL: low-density lipoprotein; HbA1c: glycated hemoglobin; TG: triglyceride; UPCR: urine protein-creatinine ratio; UACR: urine albumin-to-creatinine ratio; SBP: systolic pressure; DBP: diastolic pressure.

**Table 2 medicina-58-00577-t002:** Relation between albuminuria levels of both groups.

Proteinuria Levels	Exercise Group (%)	Non-Exercise Group (%)	*p*
Normoalbuminuria (<20 mg/L)	74 (69.8)	66 (53.7)	<0.05 **
Microalbuminuria (20–200 mg/L)	23 (21.7)	38 (30.9)	
Macroalbuminuria (>200 mg/L)	9 (8.5)	19 (15.4)	

** Difference between the exercise and non-exercise groups (*p* < 0.05).

**Table 3 medicina-58-00577-t003:** The comparison of biochemical variables between exercise and non-exercise groups stratified by three levels of normo-, micro-, and macroalbuminuria.

Variables	Normoalbuminuria (*N* = 140)			Microalbuminuria (*N* = 61)			Macroalbuminuria (*N* = 28)		
	Exercise (*n* = 74)	Non-Exercise (*n* = 66)	*p*	Exercise (*n* = 23)	Non-Exercise (*n* = 38)	*p*	Exercise (*n* = 9)	Non-Exercise (*n* = 19)	*p*
HbA1c (%)	6.81 ± 0.70	7.28 ± 1.06	<0.05 **	7.04 ± 0.68	6.78 ± 0.99	0.452	7.49 ± 1.18	8.38 ± 1.57	0.169
FPG (mg/dL)	121.6 ± 26.8	143.0 ± 45.9	<0.05 **	124.3± 18.1	129.3 ± 26.7	0.592	111.7 ± 53.0	154.4 ± 80.2	0.186
Cholesterol (mg/dL)	158.2 ± 27.6	157.1 ± 31.7	0.834	158.4 ± 25.8	157.5 ± 34.8	0.942	161.1 ± 40.6	178.8 ± 54.4	0.426
TG (mg/dL)	116.4 ± 54.9	132.8 ± 73.9	0.145	110.8 ± 46.1	169.2 ±28.6	0.148	105.8 ± 49.6	242.8 ± 214.8	<0.05 **
Uric acid (mg/dL)	7.75 ± 18.3	5.77 ± 1.33	0.422	5.63 ± 1.17	5.60 ± 1.49	0.937	5.90 ± 1.78	5.96 ± 1.96	0.943
Creatinine (mg/dL)	0.83 ± 0.18	0.80 ± 0.19	0.327	0.69 ± 0.16	0.79 ± 0.17	0.101	0.91 ± 0.15	0.89 ± 0.15	0.827
eGFR (mg/dL)	89.7 ± 18.2	93.7 ± 21.4	0.242	100.7 ± 27.9	92.9 ± 20.1	0.385	79.9 ± 12.4	81.5 ± 14.6	0.800
HDL (mg/dL)	49.8 ± 11.6	43.9 ± 8.93	<0.05 **	53.1 ± 10.4	45.7 ± 9.94	0.064 *	52.1 ± 10.3	43.3 ± 10.7	0.064 *
LDL (mg/dL)	80.9 ± 21.8	84.4 ± 23.1	0.366	77.7 ± 21.0	81.8 ± 26.8	0.667	80.3 ± 24.2	87.0 ± 29.5	0.580

* Difference between the exercise and non-exercise groups (*p* < 0.1). ** Difference between the exercise and non-exercise groups (*p* < 0.05). Abbreviations: eGFR: estimated glomerular filtration rate; FPG: fasting plasma glucose; HbA1c: glycated hemoglobin; HDL: high-density lipoprotein; LDL: low-density lipoprotein; TG: triglyceride.

**Table 4 medicina-58-00577-t004:** Multiple logistic regression analysis using exercise as a dependent variable.

Variables	Normoalbuminuria (*N* = 140)		Microalbuminuria (*N* = 61)		Macroalbuminuria (*N* = 28)	
	OR (95% CI)	*p*	OR (95% CI)	*p*	OR (95% CI)	*p*
HbA1c (%)	0.85 (0.50–1.45)	0.558	1.25 (0.61–2.58)	0.543	0.72 (0.36–1.45)	0.718
FPG (mg/dL)	0.99 (0.97–1.00)	0.061 *	0.99 (0.98–1.01)	0.312	0.99 (0.98–1.01)	0.312
TG (mg/dL)	1.00 (0.99–1.01)	0.779	1.00 (0.99–1.01)	0.203	1.02 (1.00–1.03)	<0.05 **
HDL (mg/dL)	1.06 (1.01–1.10)	<0.05 **	1.05 (0.99–1.12)	0.133	1.24 (1.08–1.44)	<0.05 **

Adjusted for gender and age. * Difference between the exercise and non-exercise groups (*p* < 0.1). ** Difference between the exercise and non-exercise groups (*p* < 0.05). Abbreviations: FPG: fasting plasma glucose; HbA1c: glycated hemoglobin; HDL: high-density lipoprotein; TG: triglyceride.

## Data Availability

The data underlying this article will be shared upon reasonable request to the corresponding author.

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
