# Peer review of "The Effects of Exercise Habit on Albuminuria and Metabolic Indices in Patients with Type 2 Diabetes Mellitus: A Cross-Sectional Study"

_medicina, 2022, doi:10.3390/medicina58050577_

Round 1
Reviewer 1 Report
Thanks for the opportunity to review the manuscript titled “The effects of exercise habit on albuminuria in patients with type 2 diabetes mellitus during the COVID-19 pandemic: A cross-sectional study”. The following are my comments .
I suggest reviewing the title, considering that the pandemic was situational and had no direct relationship with the assessment perfomed.
I suggest a better description from the methods section:
1. What were the study inclusion and exclusion criteria? How were the participants identified and how were the inclusion procedures in the study?
2. How were physical activity habits assessed? Was it considered self-report? I suggest a better description.
3. Was the presence of other diseases or medication use that could affect the albuminuria level evaluated?
4. Was sample size calculation performed to determine the power of the study?
5. Have the data been tested for normality to determine the most appropriate statistical strategy?
6. Were regression models used to correct for potential confounders?
In the results section, the authors should describe Table 1 based on the characteristics of each of the groups, in order to identify whether the groups are comparable.
Furthermore, the groups differed significantly in terms of glycated hemoglobin levels. Considering that glycemic control itself directly influences albuminuria, it is essential that the results are corrected using regression models for the participants' HbA1c.
In the discussion, I suggest reviewing the importance given to the pandemic in the evaluation carried out, considering that both groups were evaluated at the same time and that, therefore, it is not possible to associate the results found with the pandemic.
In the discussion, I also suggest revising the used references for more up-to-date references (eg, Standards of Medican Care (ADA) 2022).
It is essential to add a paragraph to the discussion presenting ALL the limitations of the study. In this paragraph, I suggest to mention the limitations inherent to the cross-sectional design, the need for an experimental study for the proposed evaluation, the lack of sample size calculation and the limitations in the statistical analysis.
Author Response
Reviewer #1:
Thanks for the opportunity to review the manuscript titled “The effects of exercise habit on albuminuria in patients with type 2 diabetes mellitus during the COVID-19 pandemic: A cross-sectional study”. The following are my comments.
Author Reply: We sincerely appreciate your time and effort spent in reviewing this manuscript. We have revised the manuscript thoroughly according to reviewer’s suggestions. The responses to your comments are found below.
I suggest reviewing the title, considering that the pandemic was situational and had no direct relationship with the assessment performed.
Author Reply: Thanks for your valuable suggestion and comments. We have made the correction immediately.
Title:
The effects of exercise habit on albuminuria and metabolic indices in patients with type 2 diabetes mellitus: A cross-sectional study
I suggest a better description from the methods section:
- What were the study inclusion and exclusion criteria? How the participants were identified and how were the inclusion procedures in the study?
Author Reply: Thanks for your valuable suggestion and comments.
- Materials and Methods
2.1. Study Subjects and Locations
A cross-sectional study was conducted by the sampling of newly diagnosed type 2 DM patients from two teaching hospitals (Taoyuan Armed Forces General Hospital and Hsin Chu Armed Hospital) in Taiwan from June 1 to December 31, 2020. All patients were participants in the Diabetes Shared Care Network in Taiwan. Demographic features, biochemical data including available blood and spot urine sample results were investigated. Including nephrologists, dietitians, and nurses were involved in this study. Only newly diagnosed type 2 DM patients on the educational program for DM and CKD with a fixed diet regimen at least one month were included. Patients with definite diagnosis of history of periodic paralysis, thyroid or adrenal disorders, inherited kidney diseases, and renal tubular acidosis were not involved in this study. These comorbidity factors and associated diseases may have potential effects on proteinuria or albuminuria. To avoid other potential confounders and focus on the target population of stable newly diagnosed type 2 DM patients, we then excluded patients with history of hypertension, acute kidney injury, massive hematuria, renal transplant, dialysis treatment, bladder irrigation, prior creation of a neobladder, pregnancy, obstructive uropathy, and age younger than 18. To avoid the effects of medications and only to investigate the original effects from exercise habit, the patients with use of any drugs including traditional Chinese medicine were also excluded in this study. For optimal health it is recommended that adults obtain at least 150 min per week of moderate-intensity. In this study, our nurses asked all the participated patients to complete the self-report of exercise habits according to the US Physical Activity Guidelines. Based on the exercise habits, the patients were divided into two groups, i.e., the exercise group (≥ 150 min per week for more than 6 months) and the non-exercise group (< 150 min per week for more than 6 months) [6, 10]. The detailed report of exercise type showed jogging mainly in all subjects.
- How was physical activity habits assessed? Was it considered self-report? I suggest a better description.
Author Reply: Thanks for your valuable suggestion and comments. We have made the correction.
- Materials and Methods
2.1. Study Subjects and Locations
A cross-sectional study was conducted by the sampling………………………
For optimal health it is recommended that adults obtain at least 150 min per week of moderate-intensity. In this study, our nurses asked all the participated patients to complete the self-report of exercise habits according to the US Physical Activity Guidelines. Based on the exercise habits, the patients were divided into two groups, i.e., the exercise group (≥ 150 min per week for more than 6 months) and the non-exercise group (< 150 min per week for more than 6 months). The detailed report of exercise type showed jogging mainly in all subjects.
- Was the presence of other diseases or medication use that could affect the albuminuria level evaluated?
Author Reply: Thanks for your valuable suggestion and comments. We have made the correction.
- Materials and Methods
2.1. Study Subjects and Locations
A cross-sectional study was conducted by the sampling………………………
Patients with definite diagnosis of history of periodic paralysis, thyroid or adrenal disorders, inherited kidney diseases, and renal tubular acidosis were not involved in this study. These comorbidity factors and associated diseases may have potential effects on proteinuria or albuminuria. To avoid other potential confounders and focus on the target population of stable newly diagnosed type 2 DM patients, we then excluded patients with history of hypertension, acute kidney injury, massive hematuria, renal transplant, dialysis treatment, bladder irrigation, prior creation of a neobladder, pregnancy, obstructive uropathy, and age younger than 18. To avoid the effects of medications and only to investigate the original effects from exercise habit, the patients with use of any drugs including traditional Chinese medicine were also excluded in this study………
- Was sample size calculation performed to determine the power of the study?
Author Reply: Thanks for your valuable suggestion and comments. We have made the correction.
2.3. Sample Size and Statistical Methods
Patients satisfying the inclusion criteria participated in the study, and the basic characteristics of demographic variables, were analysed according to category. We designed a sample size of at least 100 per each group was calculated to provide 85.4%, 80% and 99.9%, 99.5% and 97.1% of power (alpha = 0.05, two-tail) on age, HbA1c and FPG, triglyceride, HDL to detect statistically significant differences between the two groups [20].
Reference
20 Calculate Sample Size Needed to Compare 2 Means: 2-Sample Equivalence.. http://powerandsamplesize.com/Calculators/Compare-2-Means/2-Sample-Non-Inferiority-or-Superiority
- Have the data been tested for normality to determine the most appropriate statistical strategy?
Author Reply: Thanks for your valuable suggestion and comments. Using Kolmogorov–Smirnov test, we confirmed the normality of distribution of HbA1c in both exercise and non-exercise groups.
2.3. Sample Size and Statistical Methods
Patients satisfying the inclusion criteria participated in the study, and the basic characteristics of demographic variables, were analysed according to category. We designed a sample size of at least 100 per each group was calculated to provide 75% power (alpha = 0.05, two-tail) to detect statistically significant differences between the two groups. Using Kolmogorov–Smirnov test, we confirmed the normality of distribution of HbA1c, FPG, TG, LDL, and HDL in both exercise and non-exercise groups.
- Were regression models used to correct for potential confounders?
Author Reply: Thanks for your valuable suggestion and comments. We have used multiple logistic regression analysis using exercise as a dependent variable (Table 3).
Table 4. Multiple logistic regression analysis using exercise as a dependent variable
|
Variables |
Normal-albuminuria (N=140) |
Micro-albuminuria (N=61) |
Macro-albuminuria (N=28) |
|||
|
OR(95% CI) |
p |
OR(95% CI) |
p |
OR(95% CI) |
p |
|
|
HbA1c (%) |
0.85(0.50-1.45) |
0.558 |
1.25(0.61-2.58) |
0.543 |
0.72(0.36-1.45) |
0.718 |
|
FPG (mg/dl) |
0.99(0.97-1.00) |
0.061* |
0.99(0.98-1.01) |
0.312 |
0.99(0.98-1.01) |
0.312 |
|
Triglycerides (mg/dl) |
1.00(0.99-1.01) |
0.779 |
1.00(0.99-1.01) |
0.203 |
1.02(1.00-1.03) |
<.05** |
|
HDL (mg/dl) |
1.06(1.01-1.10) |
<.05** |
1.05(0.99-1.12) |
0.133 |
1.24(1.08-1.44) |
<.05** |
*Different from exercise group at 0.1 ** Different from exercise group at 0.05.
Abbreviations: HDL: high-density lipoprotein; LDL: low-density lipoprotein; HbA1c: glycated haemoglobin; TG: triglycerides
Adjusted gender and age.
In the results section, the authors should describe Table 1 based on the characteristics of each of the groups, in order to identify whether the groups are comparable.
Author Reply: Thanks for your valuable suggestion and comments. We have made the changes in Table 1.
Table 1. Demographic and Biochemical variables of the groups
|
Variables |
Exercise group (n=106) |
Non-exercise group (n=123) |
p |
|
Gender |
|
|
0.931 |
|
Female |
61(57.5) |
69(56.1) |
|
|
Male |
45(42.5) |
54(43.9) |
|
|
Age (years) |
66.2±10.1 |
61.9±12.8 |
<.05** |
|
HbA1c (%) |
6.89± 0.69 |
7.16 ± 1.05 |
<.05** |
|
FPG (mg/dl) |
121.9 ± 25.7 |
140.5 ± 42.4 |
<.05** |
|
Cholesterol (mg/dl) |
158.3 ± 27.2 |
158.5 ± 33.5 |
0.967 |
|
TG (mg/dl) |
115.6 ± 53.6 |
150.2 ± 15.4 |
<.05** |
|
Creatinine (mg/dl) |
0.81 ± 0.19 |
0.79 ± 0.19 |
0.628 |
|
eGFR (mg/dl) |
91.1 ± 19.9 |
93.6 ± 20.8 |
0.433 |
|
Uric acid (mg/dl) |
5.59 ± 1.35 |
5.85 ± 1.75 |
0.205 |
|
HDL (mg/dl) |
50.3 ± 11.4 |
44.1 ± 9.26 |
<.05** |
|
LDL (mg/dl) |
80.4 ± 21.6 |
84.1 ± 25.3 |
0.313 |
|
UPCR (mg/gm) |
199.7 ± 36.4 |
244.6 ± 34.2 |
0.371 |
|
UACR (mg/gm) |
7.66 ± 2.54 |
11.4 ± 2.41 |
0.285 |
|
SBP (mmHg) |
130.9 ± 1.36 |
130.8 ± 1.31 |
0.955 |
|
DBP (mmHg) |
76.8 ± 1.08 |
78.5 ± 1.18 |
0.287 |
|
Urine albumin (mg/L) |
63.6 ± 196.0 |
112.0 ± 291.3 |
0.139 |
Furthermore, the groups differed significantly in terms of glycated hemoglobin levels. Considering that glycemic control itself directly influences albuminuria, it is essential that the results are corrected using regression models for the participants' HbA1c.
Author Reply: Thanks for your valuable suggestion and comments.
in terms of the severity of albuminuria, the analysis of biochemical results showed that albuminuria was positively correlated with HbA1c, FPG, TG and HDL (Table 3). After multiple logistic regression analysis using exercise as a dependent variable, the analysis results also demonstrated that albuminuria was positively correlated with HbA1c, TG and HDL (Table 4).
Table 4. Multiple logistic regression analysis using exercise as a dependent variable
|
Variables |
Normal-albuminuria (N=140) |
Micro-albuminuria (N=61) |
Macro-albuminuria (N=28) |
|||
|
OR(95% CI) |
p |
OR(95% CI) |
p |
OR(95% CI) |
p |
|
|
HbA1c (%) |
0.85(0.50-1.45) |
0.558 |
1.25(0.61-2.58) |
0.543 |
0.72(0.36-1.45) |
0.718 |
|
FPG (mg/dl) |
0.99(0.97-1.00) |
0.061* |
0.99(0.98-1.01) |
0.312 |
0.99(0.98-1.01) |
0.312 |
|
Triglycerides (mg/dl) |
1.00(0.99-1.01) |
0.779 |
1.00(0.99-1.01) |
0.203 |
1.02(1.00-1.03) |
<.05** |
|
HDL (mg/dl) |
1.06(1.01-1.10) |
<.05** |
1.05(0.99-1.12) |
0.133 |
1.24(1.08-1.44) |
<.05** |
*Different from exercise group at 0.1 ** Different from exercise group at 0.05.
Adjusted gender and age.
In the discussion, I suggest reviewing the importance given to the pandemic in the evaluation carried out, considering that both groups were evaluated at the same time and that, therefore, it is not possible to associate the results found with the pandemic.
Author Reply: Thanks for your valuable suggestion and comments. We deleted all the discussions about COVID-19 pandemic in our manuscript.
- Discussion
Our study collected data from 229 newly diagnosed type 2 DM patients in Taiwan and found that HbA1c, TG and HDL were significantly different in the exercise group compared with non-exercise group. Therefore, it is speculated that an exercise duration ≥ 150 min can delay the occurrence of albuminuria and reduce the risk of DN in DM patients………
In the discussion, I also suggest revising the used references for more up-to-date references (eg, Standards of Medican Care (ADA) 2022).
Author Reply: Thanks for your valuable suggestion and comments. We have added the references (eg, Standards of Medican Care (ADA) 2021, 2022).
References
- American Diabetes Association Professional Practice Committee. 17. Diabetes Advocacy: Standards of Medical Care in Diabetes-2022. Diabetes Care 2022, 45(Suppl 1), S254-S255. doi: 10.2337/dc22-S017.
- American Diabetes Association. 2. Classification and Diagnosis of Diabetes: Standards of Medical Care in Diabetes-2021. Diabetes Care 2021, 44(Suppl 1), S15-S33. doi: 10.2337/dc21-S002.
- American Diabetes Association Professional Practice Committee; American Diabetes Association Professional Practice Committee:, Draznin, B., Aroda, V.-R., Bakris, G., Benson, G., Brown. F.-M., Freeman, R., Green, J., Huang, E., Isaacs, D., Kahan, S., Leon. J., Lyons. S.-K., Peters, A.-L., Prahalad, P., Reusch, J.-E.-B., Young-Hyman, D., Das, S., Kosiborod, M. 2. Classification and Diagnosis of Diabetes: Standards of Medical Care in Diabetes-2022. Diabetes Care. 2022, 45(Suppl 1), S17-S38. doi: 10.2337/dc22-S002.
It is essential to add a paragraph to the discussion presenting ALL the limitations of the study. In this paragraph, I suggest to mention the limitations inherent to the cross-sectional design, the need for an experimental study for the proposed evaluation, the lack of sample size calculation and the limitations in the statistical analysis.
Author Reply: Thanks for your valuable suggestion and comments. We have added the limitations.
These results of our study can serve as a basis for future research on the exercise habits of type 2 DM patients and as a reference for medical staff to educate patients in clinical practice to promote patients’ health and improve their quality of life. However, our study has several limitations: the study results may have been influenced by several physiologic factors including age, gender, nutrition, exercise type, life style, diet, and various genetic conditions. In addition, we do not collect the 24-hour urine samples to assess urinary albumin excretion in this study. Including the cross-sectional design, the need for further experimental study for the proposed evaluation, the lack of large sample size calculation and the restrictions in the statistical analysis are the limitations in this study.
Last, we sincerely appreciate your time and effort spent in reviewing this manuscript. We are motivated to read more and, thus, learn more from your criticisms.
Reviewer 2 Report
The title is misleading. Why is only albuminuria specifically written in the title
Is this study is regional specific? Why is only data from Taiwan presented in the introduction
Many sentences without any coherence are written in the introduction
The rationale of the study is not written in the introduction
Methods:
What are the inclusion criteria?
It is written as the participants were newly diagnosed DM patients. It is a vague sentence
How did the exercise time were calculated?
What about exercise intensity and type of exercise. these are the most important factors which influence which is not mentioned in the methodology
Results:
In the result section it is mentioned as – ‘the exercise group, HbA1c, FPG, BUN, TG and HDL were all significantly im-157 proved. Compared with the non-exercise group, in the exercise group, HbA1c was signif-158 icantly improved (6.89 ± 0.69 vs. 7.16 ± 1.05 %) (p < .05), FPG was significantly improved 159 (121.9 ± 25.7 vs. 140.5 ± 42.4 mg/dl) (p < .05), BUN was significantly improved (15.0 ± 4.22 160 vs. 17.3 ± 5.63 mg/dl) (p < .05), TG was significantly improved (115.6 ± 53.6 vs. 150.2 ± 115.4 161 mg/dl) (p < .05), and HDL was significantly improved,
How can the authors say that these variables are improved in a cross-sectional study?
The result section should include the results of the study, not the discussion of the results
It is written in the results section as - In terms of the sever-178 ity of albuminuria, the analysis of biochemical results showed that urine albumin was 179 positively correlated with HbA1c, AC, TG and HDL (Table 4).- How the correlation was done? What is the correlation coeff. Value? Which statistical test was performed?
The title of table 4 is misleading
Discussion :
Discussion is not according to the standard format
‘Our study collected data 192 from 229 newly diagnosed type 2 DM patients and found that HbA1c, TG and HDL were 193 significantly improved in the exercise group compared with control group during the 194 COVID-19 pandemic in Taiwan’.-Is your study is an experimental study?
‘According to the abovementioned literature, the possible reasons for the insignificant 220 improvements in total cholesterol and LDL in this study are as follows: the exercise was mainly aerobic exercise (walking, hiking, swimming, etc.). ‘ – How the author knows that exercises are aerobic exercise?
‘To the best of our knowledge, this is the first 290 report to investigate the impact of COVID-19 on the exercise habit and glycemic control’ – misleading statement – is your study investigating exercise habits?
Author Response
Reviewer #2:
The title is misleading. Why is only albuminuria specifically written in the title.
Author Reply: Thanks for your valuable suggestion and comments. We have made the correction.
Title:
The effects of exercise habit on albuminuria and metabolic indices in patients with type 2 diabetes mellitus: A cross-sectional study
Is this study regional specific? Why is only data from Taiwan presented in the introduction.
Author Reply: Thanks for your valuable suggestion and comments. We have made the correction.
- Introduction
Diabetes mellitus (DM) is a fastest increasing chronic diseases globally. The number of DM patients has quadrupled in the past three decades, and DM is the ninth major cause of death in the world. DM can cause high complications and mortality rate, then resulting in a heavy burden of medical expenses [1-3]. To date, about 1 in 11 adults worldwide have DM, 90% of whom have type 2 DM. Asia is a major area of the rapidly emerging type 2 DM global epidemic [2-5]. Even genetic predisposition could partly contribute individual susceptibility to type 2 DM, the unhealthy diet and sedentary lifestyle may play an important role in the current global epidemic [6-8]. DM is a metabolic disease and associated with chronic inflammation in a complex immunological process. Insulin resistance (IR) can also cause a series of immune responses that aggravate the inflammatory state, further resulting in hyperglycemia [9]…………..
References
- Saeedi, P., Petersohn, I., Salpea, P., Malanda, B., Karuranga, S., Unwin, N., Colagiuri, S., Guariguata, L., Motala, A.-A., Ogurtsova, K., Shaw, J.E., Bright. D., Williams, R.; IDF Diabetes Atlas Committee. Global and regional diabetes prevalence estimates for 2019 and projections for 2030 and 2045: Results from the International Diabetes Federation Diabetes Atlas, 9th edition. Diabetes Res Clin Pract 2019, 157, 107843. doi: 10.1016/j.diabres.2019.107843.
- Xu, Y.; Wang, L.; He, J.; Bi, Y.; Li, M.; Wang, T.; Wang, L.; Jiang, Y.; Dai, M.; Lu, J.; et al. Prevalence and control of diabetes in Chinese adults. JAMA 2013, 310, 948–959; DOI:10.1001/jama.2013.168118.
- Nathan, D.M.; DCCT/EDIC Research Group. The diabetes control and complications trial/epidemiology of diabetes interventions and complications study at 30 years: overview. Diabetes Care 2014, 37, 9-16. doi: 10.2337/dc13-2112.
- Zheng, Y., Ley, S.-H., Hu, F.-B. Global aetiology and epidemiology of type 2 diabetes mellitus and its complications. Nat Rev Endocrinol 2018, 14, 88-98. doi: 10.1038/nrendo.2017.151.
- Ahlqvist, E., Prasad, R.-B., Groop, L. 100 YEARS OF INSULIN: Towards improved precision and a new classification of diabetes mellitus. J Endocrinol 2021, 252, R59-R70. doi: 10.1530/JOE-20-0596.
- Lee-Ødegård, S.; Olsen, T.; Norheim, F.; Drevon, C.A.; Birkeland, K.I. Potential Mechanisms for How Long-Term Physical Activity May Reduce Insulin Resistance. Metabolites2022, 12, 208. doi: 10.3390/metabo1203020
- American Diabetes Association Professional Practice Committee. 17. Diabetes Advocacy: Standards of Medical Care in Diabetes-2022. Diabetes Care 2022, 45(Suppl 1), S254-S255. doi: 10.2337/dc22-S017.
Many sentences without any coherence are written in the introduction.
Author Reply: Thanks for your valuable suggestion and comments.
- Introduction
Diabetes mellitus (DM) is a fastest increasing chronic diseases globally. The number of DM patients has quadrupled in the past three decades, and DM is the ninth major cause of death in the world. DM can cause high complications and mortality rate, then resulting in a heavy burden of medical expenses [1-3]. To date, about 1 in 11 adults worldwide have DM, 90% of whom have type 2 DM. Asia is a major area of the rapidly emerging type 2 DM global epidemic [2-5]. Even genetic predisposition could partly contribute individual susceptibility to type 2 DM, the unhealthy diet and sedentary lifestyle may play an important role in the current global epidemic [6-8]. DM is a metabolic disease and associated with chronic inflammation in a complex immunological process. Insulin resistance (IR) can also cause a series of immune responses that aggravate the inflammatory state, further resulting in hyperglycemia [9]. Diabetic kidney disease (DKD) is one of the most common complications of DM. The pathological mechanism of DM is the lack of insulin secretion (damage to the number and/or function of β cells) and IR in liver, muscle and fat cells, which can increase blood glucose and affect lipid metabolism, resulting in increases in cholesterol and triglycerides (TG) [10,11], which further leads to arteriosclerosis and cardiovascular diseases [12]. Increases in blood glucose can also cause thickening or nodular sclerosis of the glomerular basement membrane (GBM), resulting in an increase in creatinine (Cr) and the production of proteinuria, which affects renal function [13]. At present, the three main measures for the treatment and care of DM are medication, diet and exercise; of these, exercise is the most economical. Regular exercise can reduce IR, increase insulin sensitivity and improve the efficiency of glucose entry into cells during exercise to reduce the concentration of glycated haemoglobin (HbA1c) and improve blood glucose levels [14]. In addition, exercise can improve plasma lipid levels, especially by reducing TG and increasing high-density lipoprotein (HDL) [15]. The studies of Stensvold et al. [16] and Larose et al. [17] also found that after interventions of aerobic, resistance and combined aerobic-resistance exercise, HbA1c, TG and HDL were significantly improved in the intervention group compared with the control group. Another study of 222,053 patients with the estimated glomerular filtration rate (eGFR) < 60 ml/min/1.73m2 analysed the patients’ health examination data and found that exercise could increase the eGFR [18]. The study of Chen et al. [19] retrospectively analysed the data of admitted patients from 2010 to 2012 and showed that exercise could delay the progression of renal dysfunction and the occurrence of proteinuria.
Overall, regular exercise is planned and repeated to improve or maintain physical health in human. Exercising habitually and frequently helps to prevent serious illnesses such as cardiovascular disease, heart disease, metabolic disease, type 2 DM, and obesity. Many cases of type 2 DM could be prevented with lifestyle modifications, such as remaining a healthy body weight, keeping on a healthy diet, maintaining a physically active, not smoking and not drinking alcohol. The aim of our present work is to explore the association between exercise habit and albuminuria and metabolic indices in patients with newly diagnosed with type 2 DM. This observational study also investigated the potential physiological effect of exercise among newly diagnosed type 2 DM patients
The rationale of the study is not written in the introduction.
Author Reply: Thanks for your valuable suggestion and comments. We have made the correction.
- Introduction
Diabetes mellitus (DM) is a fastest increasing chronic diseases globally. The number of DM patients has quadrupled in the past three decades, and DM is the ninth major cause of death in the world. DM can cause high complications and mortality rate, then resulting in a heavy burden of medical expenses [1-3]. To date, about 1 in 11 adults worldwide have DM, 90% of whom have type 2 DM. Asia is a major area of the rapidly emerging type 2 DM global epidemic [2-5]. Even genetic predisposition could partly contribute individual susceptibility to type 2 DM, the unhealthy diet and sedentary lifestyle may play an important role in the current global epidemic [6-9]. Diabetic kidney disease (DKD) is one of the most common complications of DM. The pathological mechanism of DM is the lack of insulin secretion (damage to the number and/or function of β cells) and insulin resistance (IR) in liver, muscle and fat cells, which can increase blood glucose and affect lipid metabolism, resulting in increases in cholesterol and triglycerides (TG) [10,11], which further leads to arteriosclerosis and cardiovascular diseases [12]. Increases in blood glucose can also cause thickening or nodular sclerosis of the glomerular basement membrane (GBM), resulting in an increase in creatinine (Cr) and the production of proteinuria, which affects renal function [13]. At present, the three main measures for the treatment and care of DM are medication, diet and exercise; of these, exercise is the most economical. Regular exercise can reduce IR, increase insulin sensitivity and improve the efficiency of glucose entry into cells during exercise to reduce the concentration of glycated haemoglobin (HbA1c) and improve blood glucose levels [14]. In addition, exercise can improve plasma lipid levels, especially by reducing TG and increasing high-density lipoprotein (HDL) [15]. The studies of Stensvold et al. [16] and Larose et al. [17] also found that after interventions of aerobic, resistance and combined aerobic-resistance exercise, HbA1c, TG and HDL were significantly improved in the intervention group compared with the control group. Another study of 222,053 patients with the estimated glomerular filtration rate (eGFR) < 60 ml/min/1.73m2 analysed the patients’ health examination data and found that exercise could increase the eGFR [18]. The study of Chen et al. [19] retrospectively analysed the data of admitted patients from 2010 to 2012 and showed that exercise could delay the progression of renal dysfunction and the occurrence of proteinuria.
Overall, regular exercise is planned and repeated to improve or maintain physical health in human. Exercising habitually and frequently helps to prevent serious illnesses such as cardiovascular disease, heart disease, metabolic disease, type 2 DM, and obesity. Many cases of type 2 DM could be prevented with lifestyle modifications, such as remaining a healthy body weight, keeping on a healthy diet, maintaining a physically active, not smoking and not drinking alcohol. The aim of our present work is to explore the association between exercise habit and albuminuria and metabolic indices in patients with newly diagnosed with type 2 DM. This observational study also investigated the potential physiological effect of exercise among newly diagnosed type 2 DM patients.
Methods:
What are the inclusion criteria?
Author Reply: Thanks for your valuable suggestion and comments. We have made the correction.
- Materials and Methods
2.1. Study Subjects and Locations
A cross-sectional study was conducted by the sampling of newly diagnosed type 2 DM patients from two teaching hospitals (Taoyuan Armed Forces General Hospital and Hsin Chu Armed Hospital) in Taiwan from June 1 to December 31, 2020. All patients were participants in the Diabetes Shared Care Network in Taiwan. Demographic features, biochemical data including available blood and spot urine sample results were investigated. Including nephrologists, dietitians, and nurses were involved in this study. Only newly diagnosed type 2 DM patients on the educational program for DM and CKD with a fixed diet regimen at least one month were included. Patients with definite diagnosis of history of periodic paralysis, thyroid or adrenal disorders, inherited kidney diseases, and renal tubular acidosis were not involved in this study. These comorbidity factors and associated diseases may have potential effects on proteinuria or albuminuria. To avoid other potential confounders and focus on the target population of stable newly diagnosed type 2 DM patients, we then excluded patients with history of hypertension, acute kidney injury, massive hematuria, renal transplant, dialysis treatment, bladder irrigation, prior creation of a neobladder, pregnancy, obstructive uropathy, and age younger than 18. To avoid the effects of medications and only to investigate the original effects from exercise habit, the patients with use of any drugs including traditional Chinese medicine were also excluded in this study.
It is written as the participants were newly diagnosed DM patients. It is a vague sentence.
Author Reply: Thanks for your valuable suggestion and comments. We have made the correction.
- Materials and Methods
2.1. Study Subjects and Locations
A cross-sectional study was conducted by the sampling of newly diagnosed type 2 DM patients from two teaching hospitals (Taoyuan Armed Forces General Hospital and Hsin Chu Armed Hospital) in Taiwan from June 1 to December 31, 2020. All patients were participants in the Diabetes Shared Care Network in Taiwan. Demographic features, biochemical data including available blood and spot urine sample results were investigated. Including nephrologists, dietitians, and nurses were involved in this study. Only newly diagnosed type 2 DM patients on the educational program for DM and CKD with a fixed diet regimen at least one month were included
How did the exercise time were calculated?
Author Reply: Thanks for your valuable suggestion and comments.
- Materials and Methods
2.1. Study Subjects and Locations
A cross-sectional study was conducted by the sampling………………………
For optimal health it is recommended that adults obtain at least 150 min per week of moderate-intensity. In this study, our nurses asked all the participated patients to complete the self-report of exercise habits according to the US Physical Activity Guidelines. Based on the exercise habits, the patients were divided into two groups, i.e., the exercise group (≥ 150 min per week for more than 6 months) and the non-exercise group (< 150 min per week for more than 6 months). The detailed report of exercise type showed jogging mainly in all subjects.
What about exercise intensity and type of exercise. These are the most important factors which influence which is not mentioned in the methodology
Author Reply: Thanks for your valuable suggestion and comments. We have made the correction. The detailed report of exercise type shows jogging mainly in all samples; this is also a limitation in our study, we have added this in the limitation.
- Materials and Methods
2.1. Study Subjects and Locations
A cross-sectional study was conducted by the sampling………………………
For optimal health it is recommended that adults obtain at least 150 min per week of moderate-intensity. In this study, our nurses asked all the participated patients to complete the self-report of exercise habits according to the US Physical Activity Guidelines. Based on the exercise habits, the patients were divided into two groups, i.e., the exercise group (≥ 150 min per week for more than 6 months) and the non-exercise group (< 150 min per week for more than 6 months). The detailed report of exercise type showed jogging mainly in all subjects.
These results of our study can serve as a basis for future research on the exercise habits of type 2 DM patients and as a reference for medical staff to educate patients in clinical practice to promote patients’ health and improve their quality of life. However, our study has several limitations: the study results may have been influenced by several physiologic factors including age, gender, nutrition, exercise type, life style, diet, and various genetic conditions. In addition, we do not collect the 24-hour urine samples to assess urinary albumin excretion in this study. Including the cross-sectional design, the need for further experimental study for the proposed evaluation, the lack of large sample size calculation and the restrictions in the statistical analysis are the limitations in this study
Results:
In the result section it is mentioned as – ‘the exercise group, HbA1c, FPG, BUN, TG and HDL were all significantly im-157 proved. Compared with the non-exercise group, in the exercise group, HbA1c was signif-158 icantly improved (6.89 ± 0.69 vs. 7.16 ± 1.05 %) (p < .05), FPG was significantly improved 159 (121.9 ± 25.7 vs. 140.5 ± 42.4 mg/dl) (p < .05), BUN was significantly improved (15.0 ± 4.22 160 vs. 17.3 ± 5.63 mg/dl) (p < .05), TG was significantly improved (115.6 ± 53.6 vs. 150.2 ± 115.4 161 mg/dl) (p < .05), and HDL was significantly improved,
The result section should include the results of the study, not the discussion of the results.
Author Reply: Thanks for your valuable suggestion and comments. We have made the correction.
- Results
A total of 240 patients were initially included in this study. To avoid the effects of drugs, we then excluded 4 and 7 patients due to the use of anti-hypertensive drugs and oral hypoglycemic agents in this study. A total of 229 patients completed this study, including 99 males (43.2%) and 132 females (57.8%) with an average age of 63.41 ± 11.81 years. A total of 106 patients comprised the exercise group (46.3%), and 123 patients comprised the non-exercise group (53.7%) (Table 1).
In the exercise group, HbA1c, FPG, BUN, TG and HDL were all significantly different. Compared with the non-exercise group, in the exercise group, HbA1c was significantly lower (6.89 ± 0.69 vs. 7.16 ± 1.05 %) (p < .05), FPG was significantly lower (121.9 ± 25.7 vs. 140.5 ± 42.4 mg/dl) (p < .05), BUN was also significantly lower (15.0 ± 4.22 vs. 17.3 ± 5.63 mg/dl) (p < .05), TG was significantly lower (115.6 ± 53.6 vs. 150.2 ± 115.4 mg/dl) (p < .05), and HDL was significantly higher (50.3 ± 11.4 vs. 44.1 ± 9.26 mg/dl) (p < .05) (Table 1).
It is written in the results section as - In terms of the severity of albuminuria, the analysis of biochemical results showed that urine albumin was positively correlated with HbA1c, AC, TG and HDL (Table 4).- How the correlation was done? What is the correlation coeff. Value? Which statistical test was performed?
Author Reply: Thanks for your valuable suggestion and comments. We have made the correction. According to your suggestion, we plus the supplementary table 1 with GLM Regression Coefficients as the following as well.
GLM Regression Coefficients table- Urine albumin (mg/L)
|
|
Unstandardized Coefficients |
Standardized Coefficients |
|
|
95% CI for B |
||
|
Model |
B |
Std. Error |
Beta |
t |
p |
Lower Bound |
Upper Bound |
|
1(Constant) |
-120.4 |
88.87 |
|
-1.36 |
0.177 |
-295.9 |
55.07 |
|
HbA1c (%) |
21.44 |
12.58 |
0.133 |
1.70 |
0.090* |
-3.42 |
46.29 |
|
1(Constant) |
|
|
|
|
|
|
|
|
FPG (mg/dl) |
0.71 |
0.31 |
0.177 |
2.27 |
<.05** |
0.092 |
1.330 |
|
1(Constant) |
-94.72 |
16.48 |
|
-5.75 |
<.05** |
-127.3 |
-62.18 |
|
Triglycerides (mg/dl) |
0.94 |
0.10 |
0.588 |
9.18 |
<.05** |
0.74 |
1.15 |
|
1 (Constant) |
132.03 |
50.51 |
|
2.61 |
<0.05** |
32.27 |
231.8 |
|
HDL (mg/dl) |
-2.17 |
1.04 |
-0.16 |
-2.08 |
<0.05** |
-4.22 |
-0.11 |
|
1 (Constant) |
-181.1 |
64.9 |
|
-2.79 |
<0.05** |
-309.3 |
-52.77 |
|
FPG (mg/dl) |
0.39 |
0.27 |
0.096 |
1.46 |
0.146 |
-0.14 |
0.91 |
|
Triglycerides (mg/dl) |
0.95 |
0.11 |
0.592 |
8.68 |
<.05** |
0.73 |
1.16 |
|
HDL (mg/dl) |
0.74 |
0.92 |
0.055 |
0.80 |
0.424 |
-1.08 |
2.56 |
|
1 (Constant) |
-173.0 |
91.0 |
|
-1.90 |
0.059 |
-352.8 |
6.74 |
|
HbA1c (%) |
7.21 |
10.5 |
0.045 |
0.69 |
0.494 |
-13.56 |
27.99 |
|
Triglycerides (mg/dl) |
0.95 |
0.11 |
0.595 |
8.66 |
<.05** |
0.74 |
1.17 |
|
HDL (mg/dl) |
0.56 |
0.92 |
0.042 |
0.61 |
0.543 |
-1.25 |
2.36 |
|
1 (Constant) |
-179.7 |
91.0 |
|
-1.98 |
0.050 |
-359.3 |
0.028 |
|
HbA1c (%) |
-0.27 |
12.0 |
-0.002 |
-0.02 |
0.982 |
-23.97 |
23.45 |
|
FPG (mg/dl) |
0.39 |
0.30 |
0.097 |
1.28 |
0.201 |
-0.21 |
0.99 |
|
Triglycerides (mg/dl) |
0.95 |
0.11 |
0.592 |
8.62 |
<.05** |
0.73 |
1.17 |
|
HDL (mg/dl) |
0.74 |
0.92 |
0.055 |
0.80 |
0.425 |
-1.09 |
2.56 |
GLM: general linear model
The title of table 4 is misleading.
Author Reply: Thanks for your valuable suggestion and comments. We have made the correction.according to your suggestion, we rearrange table 4 as table 3 which tile is amended as the following as.
Table 3 The comparison of biochemical variables between exercise and non-exercise groups stratified via 3 levels with normal-, micro- and macro- albuminuria
How can the authors say that these variables are improved in a cross-sectional study?
Author Reply: Thanks for your valuable suggestion and comments. We have made all the corrections in the result section.
- Results
A total of 240 patients were initially included in this study. To avoid the effects of drugs, we then excluded 4 and 7 patients due to the use of anti-hypertensive drugs and oral hypoglycemic agents in this study. A total of 229 patients completed this study, including 99 males (43.2%) and 132 females (57.8%) with an average age of 63.41 ± 11.81 years. A total of 106 patients comprised the exercise group (46.3%), and 123 patients comprised the non-exercise group (53.7%) (Table 1).
In the exercise group, HbA1c, FPG, TG and HDL were all significantly different. Compared with the non-exercise group, study results showed lower HbA1c (6.89 ± 0.69 vs. 7.16 ± 1.05 %) (p < .05), lower FPG (121.9 ± 25.7 vs. 140.5 ± 42.4 mg/dl) (p < .05), lower TG (115.6 ± 53.6 vs. 150.2 ± 115.4 mg/dl) (p < .05), and higher HDL (50.3 ± 11.4 vs. 44.1 ± 9.26 mg/dl) (p < .05) levels were noted in the exercise group (Table 1).
This study found that patients with normal albuminuria were predominant in both groups, and there were more patients with normal albuminuria in the exercise group than in the non-exercise group (69.8% vs. 53.7%). There were fewer patients with microalbuminuria and macroproteinuria in the exercise group than in the non-exercise group (21.7% vs. 30.9% and 8.5% vs. 15.4%) (Table 2). In the exercise group, there were fewer patients with microalbuminuria or macroalbuminuria (Figure 1).
In terms of the severity of albuminuria, the analysis of biochemical results showed that albuminuria was significantly correlated with HbA1c, FPG, TG and HDL (Table 3). After multiple logistic regression analysis using exercise as a dependent variable, the analysis results also demonstrated a trend that albuminuria was strongly correlated with HbA1c, TG and HDL (Table 4). Using general linear model (GLM) with regression coefficients was shown in the Supplementary Table 1.
Discussion :
Discussion is not according to the standard format
‘Our study collected data from newly diagnosed type 2 DM patients and found that HbA1c, TG and HDL were significantly improved in the exercise group compared with control group during the COVID-19 pandemic in Taiwan’.-Is your study is an experimental study?
Author Reply: Thanks for your valuable suggestion and comments. We have made the correction.
- Discussion
Our study collected data from 229 newly diagnosed type 2 DM patients in Taiwan and found that HbA1c, TG and HDL were significantly different in the exercise group compared with non-exercise group. Therefore, it is speculated that an exercise duration ≥ 150 min can delay the occurrence of albuminuria and reduce the risk of DN in DM patients……..
‘To the best of our knowledge, this is the first 290 report to investigate the impact of COVID-19 on the exercise habit and glycemic control’ – misleading statement – is your study investigating exercise habits?
Author Reply: Thanks for your valuable suggestion and comments. We deleted the sentence and made the correction.
Last, we sincerely appreciate your time and effort spent in reviewing this manuscript. We are motivated to read more and, thus, learn more from your criticisms.
Reviewer 3 Report
The authors studied the effects of regular exercise in DM patients during the pandemic and found that the metabolic status of those who exercised more than 150 hours per week the past 6 months was better compared to the other group. This was a cross sectional study.
I feel that the manuscript requires a major revision and focus must be given to a shorter introduction and discussion as well as improved results paragraph. Moreover, a description of the exercise should be given in description of the material and methods and not in the discussion. This applies to other aspects of the manuscript as well; for instance any discussion of the results should be kept in the discussion and not in the results.
Author Response
Reviewer #3:
The authors studied the effects of regular exercise in DM patients during the pandemic and found that the metabolic status of those who exercised more than 150 hours per week the past 6 months was better compared to the other group. This was a cross sectional study.
Author Reply: We sincerely appreciate your time and effort spent in reviewing this manuscript. We have revised the manuscript thoroughly according to your suggestions. The responses to your comments are found below.
I feel that the manuscript requires a major revision and focus must be given to a shorter introduction and discussion as well as improved results paragraph. Moreover, a description of the exercise should be given in description of the material and methods and not in the discussion. This applies to other aspects of the manuscript as well; for instance any discussion of the results should be kept in the discussion and not in the results.
Author Reply: Thanks for your valuable suggestion and comments. We have made all the corrections to you and previous 2 reviewers.
- Results
A total of 240 patients were initially included in this study. To avoid the effects of drugs, we then excluded 4 and 7 patients due to the use of anti-hypertensive drugs and oral hypoglycemic agents in this study. A total of 229 patients completed this study, including 99 males (43.2%) and 132 females (57.8%) with an average age of 63.41 ± 11.81 years. A total of 106 patients comprised the exercise group (46.3%), and 123 patients comprised the non-exercise group (53.7%) (Table 1).
In the exercise group, HbA1c, FPG, TG and HDL were all significantly different. Compared with the non-exercise group, study results showed lower HbA1c (6.89 ± 0.69 vs. 7.16 ± 1.05 %) (p < .05), lower FPG (121.9 ± 25.7 vs. 140.5 ± 42.4 mg/dl) (p < .05), lower TG (115.6 ± 53.6 vs. 150.2 ± 115.4 mg/dl) (p < .05), and higher HDL (50.3 ± 11.4 vs. 44.1 ± 9.26 mg/dl) (p < .05) levels were noted in the exercise group (Table 1).
This study found that patients with normal albuminuria were predominant in both groups, and there were more patients with normal albuminuria in the exercise group than in the non-exercise group (69.8% vs. 53.7%). There were fewer patients with microalbuminuria and macroproteinuria in the exercise group than in the non-exercise group (21.7% vs. 30.9% and 8.5% vs. 15.4%) (Table 2). In the exercise group, there were fewer patients with microalbuminuria or macroalbuminuria (Figure 1).
In terms of the severity of albuminuria, the analysis of biochemical results showed that albuminuria was significantly correlated with HbA1c, FPG, TG and HDL (Table 3). After multiple logistic regression analysis using exercise as a dependent variable, the analysis results also demonstrated a trend that albuminuria was strongly correlated with HbA1c, TG and HDL (Table 4). Using general linear model (GLM) with regression coefficients was shown in the Supplementary Table 1.
- Discussion
Our study collected data from 229 newly diagnosed type 2 DM patients in Taiwan and found that HbA1c, TG and HDL were significantly different in the exercise group compared with non-exercise group. Therefore, it is speculated that an exercise duration ≥ 150 min can delay the occurrence of albuminuria and reduce the risk of DN in DM patients.
The regular exercise is an important tool in lifestyle because it could reduce cardiovascular disease, DM, and malignancy and associated with decreasing hypertension, overweight, and obesity. In the past, Stensvold et al. [16] and Larose et al. [17] recommended that interventions of aerobic, resistance and combined aerobic-resistance exercise may significantly improve levels of HbA1c, TG and HDL in patients with metabolic syndrome. The study of Lin et al. [21] indicated that exercise can reduce total cholesterol and LDL, while in this study, total cholesterol and LDL were not significantly improved. According to the literature, DM patients should consider the type, intensity, frequency, duration of exercise. Types of exercise include the following: (1) aerobic exercise, (2) muscular endurance exercise, (3) stretching exercises and (4) balance exercises. The exercise intensity should be low or moderate. Both the rating of perceived exertion (RPE) and the targeted heart rate (THR) = maximum heart rate (220-age) x50 -70% recommended by the 2017 American Diabetes Association (ADA): Standards of Medical Care can be used as references, and the most suitable exercise intensity for DM patients is one at which the patient can talk freely but cannot sing freely (i.e., an intensity that causes them to breathe heavily) during exercise. The recommended exercise frequency is at least 150 min per week [22]. Exercise should take place within 1-2 hours after a meal and can take place multiple times per day, and each exercise session should be longer than 20 min [23].
Although the exercise duration was more than 150 min per week, whether the exercise intensity met the RPE recommendations of the 2022 ADA Standards of Medical Care was not evaluated in this study [24]. In addition, muscular endurance exercises with dumbbells, extendible ropes and resistance bands, stretching exercises and balance exercises, such as walking in a straight line, can be included to reduce joint stiffness and the risk of falls [18]. Another study found that exercise can improve BP [25,26], but the present study showed no significant difference in BP between the two groups. In addition to exercise, diet and medicines also affect BP. The BP of the two groups was controlled at 130 ± 1.3/80 ± 1.1 mmHg, which is close to the ideal BP range.
Renal function indicators, such as Cr, eGFR, UPCR and UACR, also showed no significant differences between the two groups. According to the literature, the following factors affect renal function: (A) Age: eGFR declines with age; (B) BP: if BP is not controlled to below 130/80 mmHg, it can cause glomerulosclerosis and affect renal function; (C) Blood glucose: the advanced glycation end-products (AGEs) and cytokines produced by hyperglycaemia (HbA1c above 7%) can cause mesangial expansion and reduce the eGFR; (D) Plasma lipids, especially LDL: if LDL is not controlled to below 100 mg/dl, renal atherosclerosis, which affects renal function, can easily occur [13,27].
Among the indicators of albuminuria, microalbuminuria is a diagnostic indicator and an early clinical manifestation of DN. DN is the most common chronic complication of DM and is one of the major causes of end-stage renal failure. Therefore, prevention of the progression of DM to DN is an important clinical task. Urinary albumin excretion, i.e., albuminuria, is a commonly used diagnostic criterion for DN. Albuminuria is divided into normal albuminuria (urine albumin < 20 mg/L), microalbuminuria (urine albumin 20-200 mg/L), and macroalbuminuria (urine albumin > 200 mg/L) [3-5]. Proteinuria or albuminuria is also an imperative marker of estimation of renal functions, which can help the early detection for progression of kidney diseases [28]. As well as being a major indication of kidney diseases, albuminuria constitutes a marker of cardiovascular disease and kidney diseases progression in addition to a predictor of death. A reduction in the degree of albuminuria has been demonstrated to improve both CV and renal outcomes [29,30]. In our study, the patients were divided by exercise duration.
Exercise training has been recommended for patients with CKD by the Kidney Disease Improving Global Outcomes [31]. A substantial number of meta-analyses confirmed the positive impacts of regular exercise programs for CKD patients on physical performance, cardiopulmonary function, blood lipids, and quality of life [32-34]. Previous review suggested that high levels of physical activity appeared to be closely related to low proteinuria [35], and an observational study of non-diabetic women had similar results [36]. Afshinnia et al. [37] recommended that exercise training can reduce proteinuria in obese people, though the long-term effect has not been evaluated by high-quality experimental studies. However, this sedentary time of CKD patients, especially those with severe renal function impairment, is still significantly higher than that of individuals without CKD. Glavinovic et al. [38] reported that the sedentary time of CKD was 10-times higher than that of individuals without CKD. Based on the results of our study, in clinical practice, health education provided by medical staff to DM patients should include the following: (1) the exercise duration should be ≥ 150 min; (2) a self-assessment of exercise intensity is the ability to talk freely but not to sing freely (i.e., breathing heavily) during exercise. With exercise that meets these indicators, blood glucose, TG and LDL can be reduced, HDL can be increased, and renal function can be maintained.
These results of our study can serve as a basis for future research on the exercise habits of type 2 DM patients and as a reference for medical staff to educate patients in clinical practice to promote patients’ health and improve their quality of life. However, our study has several limitations: the study results may have been influenced by several physiologic factors including age, gender, nutrition, exercise type (walking, hiking, swimming, etc), life style, diet, and various genetic conditions. In addition, we do not collect the 24-hour urine samples to assess urinary albumin excretion in this study. Including the cross-sectional design, the need for further experimental study for the proposed evaluation, the lack of large sample size calculation and the restrictions in the statistical analysis are the limitations in this study.
Last, we sincerely appreciate your time and effort spent in reviewing this manuscript. We are motivated to read more and, thus, learn more from your criticisms.
Round 2
Reviewer 1 Report
The authors made all the suggested modifications. No new changes to suggest.
Reviewer 2 Report
The paper has been improved. It can be accepted
Reviewer 3 Report
Thank you for your substantial revision. In my opinion the aims, methods and results have been a lot more clarified and the manuscript more robust.
I have no more comments.